# Role of Water Policies in the Adoption of Smart Water Metering and the Future Market



**Spancer Msamadya** [1], **Jin Chul Joo** [1,*], **Jung Min Lee** [2,*], **Jong Soo Choi** [2], **Sangho Lee** [3], **Doo Jin Lee** [4], **Hyeon Woo Go** [1], **So Ye Jang** [1] and **Dong Hwi Lee** [1]

1    Department of Civil and Environmental Engineering, Hanbat National University, Daejeon 34158, Korea; msamadya@gmail.com (S.M.); rhgusdn806@naver.com (H.W.G.); jjscat123@naver.com (S.Y.J.); sp3836@naver.com (D.H.L.)
2    Department of Construction Environment Research, Land and Housing Institute, Daejeon 34047, Korea; jongsoo@lh.or.kr
3    School of Civil and Environmental Engineering, Kookmin University, Seoul 02707, Korea; sanghlee@kookmin.ac.kr
4    Korea Water Resources Corporation (K-Water), Daejeon 34350, Korea; djlee@kwater.or.kr
*    Correspondence: jincjoo@hanbat.ac.kr (J.C.J.); andrew4502@lh.or.kr (J.M.L.); Tel.: +82-42-821-1264 (J.C.J.); +82-42-866-8464 (J.M.L.)

**Abstract:** Both status and progress in smart water metering (SWM) implementations in five selected countries (i.e., United States of America, United Kingdom, Australia, Israel, and South Korea) are investigated in this study. Despite the countless benefits of SWM implementation, the diffusion of the SWM technologies has been slow due to various challenges, including the absence of compulsory water policies, the lack of support from customers and expertise, and weak cost–benefit analysis. Over the past 30 years, the aforementioned countries have transitioned from a fixed charging to a volumetric charging regime composed of traditional water meters and SWM. Both the status and progress of SWM implementation are quite different among countries, although governments across the world have been applying water policies responding to water scarcity, population growth, and water demand management. However, the absence of strong water policies and political support for SWM implementation resulted in the slow and retarded spread of SWM implementation. Although several changes in water policies have occurred since 1990, there is no compulsory law for SWM implementation. Between 1995 and 2010, pilot/trial cases for SWM were dominant. After 2010, the number of SWM implementation kept increasing and all countries experienced more concentrated SWM implementation, despite the variances in both endpoints and completion of SWM implementation depending on water policies (i.e., acts and regulations) encouraging SWM implementations. The global market for SWM has consistently grown to USD 5.92 billion in 2020. Finally, the application of favourable water policies to optimize the use of water resources and to promote sustainable development is expected to drive the SWM market further.

**Keywords:** implementation; pilot/trial cases; smart water metering (SWM); traditional water meters; water policies

## 1. Introduction

According to world population prospects by the United Nations, the world's population is expected to reach 9.8 billion by 2050 [1]. With population growth and economic development, global water demand is expected to increase significantly and more than 40% of the global population can be under severe water stress by 2050 [2]. This population growth and climate change will pose substantial challenges for water management and increase the pressure of water supplies, resulting in threats to urban water security in water-scarce cities [3,4]. For example, various water utilities are struggling to maintain a stable water supply during water shortages or in peak demand times without real-time

water usage information and measures [5,6]. Thus, real-time water consumption data are required to identify water usage and to enhance the efficiency of water utilities [7,8].

To effectively manage water supply and demand, numerous water utilities have adopted smart water metering (SWM) [9] and SWM has an important role to measure the amount of water supplied to a consumer over a specified period for billing purposes, to provide high resolution and frequent water consumption data, and to enhance water conservation and management [10]. In recent decades, demand-side water management has emerged to complement traditional supply side operations that foster water conservation and more efficient water demands [11]. Additionally, the rise of demand-side water management has motivated the development of more sophisticated SWM technologies to monitor, characterize, and predict water demands at different spatial and temporal scales [12,13].

In general, SWM has been reported to improve operations for stable water supply through reducing water consumption per capita, leakages, and operational and maintenance costs [14,15]. The major benefits of SWM have been reported as follows: (1) monitor the flow, distribution, and consumption of water; (2) enable real-time access to water consumption information and billing; (3) increase water use efficiency and conservation; (4) improve leak and fraud detection; and (5) provide more accurate readings and data collections [16].

For most countries, small-scale SWM pilot/trial cases have been followed by large-scale SWM implementations. Most SWM projects from SWM pilot/trial cases to actual implementations are residentially focused; however, some SWM projects are industrial and commercially focused [17,18]. Additionally, most SWM implementations have occurred in North America (e.g., USA and Canada), Europe (e.g., Germany, France, and UK), Middle East (e.g., Israel, Saudi Arabia, and United Arab Emirates), and Asia (e.g., Australia, China, Japan, and South Korea). Therefore, in this study, a comprehensive snapshot of both the status and progress of SWM implementations from selected countries is investigated to explore the main drivers supporting the adoption of SWM technologies. Thus, both transition period and water policies (i.e., laws and regulations) shifting from pilot/trial cases to actual SWM implementations are intensively investigated in several selected countries (i.e., USA, UK, Australia, Israel, and South Korea). This review paper mainly analyses both status and progress in SWM implementation in the selected countries and focuses on reviewing the transition from pilot/trial cases to SWM implementation. Data are collected by reviewing water policies, journals, news articles, utility websites, and close communication with stakeholders involved. The specific objectives of this study are to (1) assess the diffusion status of SWM implementation in several selected countries, (2) evaluate the impact of water policies on water metering in selected countries, (3) compare the water policies supporting SWM implementation in selected countries, and (4) asses the historical market trend and predict the future sales of SWM implementation in the selected countries.

## 2. Smart Water Metering: Benefits and Challenges

Recently, smart water metering (SWM) has emerged as a vital tool in the application of water demand management in response to a range of variables, including socio-demographic factors, contextual factors, and external and internal factors [19–21]. Thus, numerous water industries and national governments have applied SWM to overcome the supply side management limitations, to assist in diagnosis, prioritization, and management of water through the water usage data, and to optimize all aspects of the water distribution network [19–21]. Although intelligent metering can be classified as automated meter reading (AMR) or advanced metering infrastructure (AMI) according to the level of sophistication of measurement and control, or operability [22], in this study, SWM is referred to as the umbrella term for AMI and AMR.

In a recent review paper [23], 75 benefits of the SWM technologies were revealed and discussed. This review attracted the input of several experts in water management through questionnaires, revealing benefits beyond metering and billing for consideration by

stakeholders intending to implement SWM. Combined with SWM capabilities, such as the remotely accessible and reliable provision, accurate water metering resulted in improved water accounting, service delivery efficiency, and compliance [23]. As acceleration of global warming, population growth and urbanization continues, which increases the pressure on the available water resources, SWM implementation can provide a solution to relieve water scarcity [2,3,23]. Liu et al. (2018) reviewed 25 studies conducted in several countries at different temporal and spatial scales and reported that water savings up to 24.1% in Sacramento County were achieved, highlighting the significant impact of SWM implementation on water consumption [17]. The aggregated individual household savings produce significant water use reductions in the supply chain, which help to reduce pressure on water resources. Finally, SWM implementation impacts residential water consumption volumes and accuracy, frequency, and remoteness of meter reading, influencing the sales of water utilities in a positively way.

Despite the countless benefits that can be derived through SWM implementation, the diffusion of SWM technologies has been slow. Various challenges to SWM implementation include the lack of support from customers, weak cost–benefit analysis, lack of expertise, and the absence of supporting water policies. Additionally, challenges regarding privacy and security, data management, technical capacity, and customer support should be overcome to ensure the widespread implementation of SWM [22]. Extra challenges regarding the higher cost of SWM technologies and the poor communication of SWM benefits also need to be overcome [5]. Furthermore, an apparent lack of standards and common protocols for SWM implementation generates more challenges for water companies, signifying that smarter systems become difficult to select or to develop, as various options may be applicable in different settings [24,25]. Particularly, the successful SWM implementation significantly depends on water policies and the decisiveness of both national government and water providers. Since government legislation and guidance have often been unclear and limited [20,26], the lack of robust water policies for SWM implementation has resulted in a slow SWM spread [10].

## 3. Implementation Status of Smart Water Metering

### 3.1. United States of America (USA)

In the USA, 41 SWM projects inclusive of pilot/trial cases and implementations were reviewed. As presented in Figure 1a, different percentages of completion for SWM implementation were observed among different cities in the USA as of 2020. Several cities (i.e., City of Columbia, Kansas City, and San Francisco) have completed SWM installation, and are now billing and obtaining the numerous SWM implementation benefits [27,28]. Several SWM implementation have been completed in the state of California (CA) especially in the period from 2015 to the present [29]. Since the state of CA experienced a 4-year severe drought from 2012 to 2016, the drought was a warning for local water authorities and the government to explore new water efficiency approaches to save every drop of precious water resources. Consequently, a drought response program by the Bureau of Reclamation and the CA government issued grants to cities in CA to assist in SWM implementation, resulting in the spread of SWM implementation throughout CA [30].

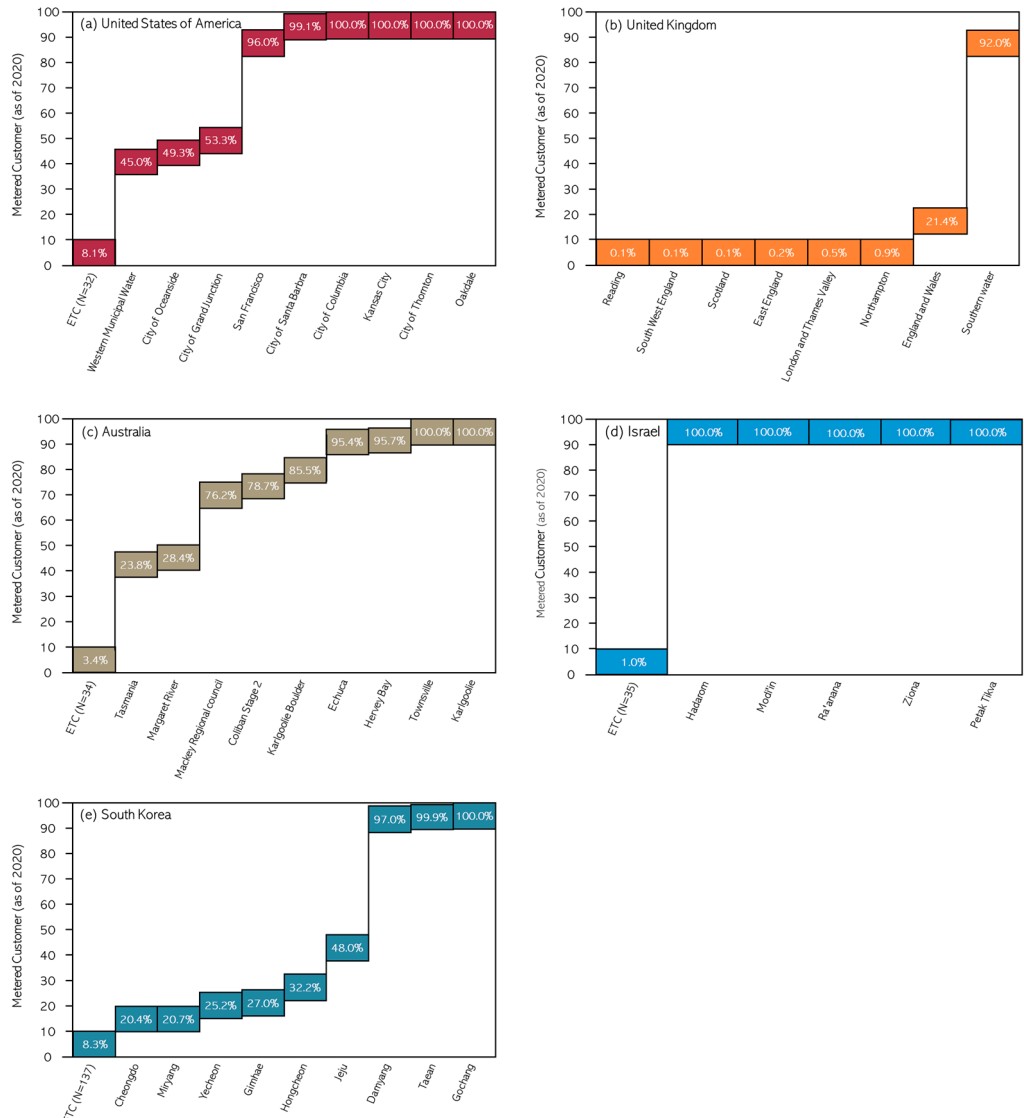

**Figure 1.** Smart water metering implementation status of several cities in (**a**) United States of America, (**b**) United Kingdom, (**c**) Australia, (**d**) Israel, and (**e**) South Korea.

In addition to cities in CA installing almost complete SWM implementation, Kansas City installed a SWM system to serve its 167,000 customers [27]. Water utilities in Kansas City managed to reduce billing costs because the geographic area of Kansas City is 318 square miles, indicating that customer field service trips were relatively expensive due to the long-distance travel for manual meter reading. The complete SWM implementation resulted in the reduction in field trips, labour costs, and customer call centre volumes [27]. Additionally, San Francisco Public Service Commission in Northern California installed 180,000 SWM, allowing customers to monitor their water usage in real time [28]. SWM implementation is being expedited in varying sizes from small rural towns with less than 1000 connections to large cities, such as New York, Boston, Cincinnati, Philadelphia, and Chicago, serving up to 1 million connections [31].

However, as is evident from Figure 1a, many miscellaneous cities with an average completion of 8.02% in SWM implementation existed as of 2020. This phenomenon may be attributed to the absence of law enforcement and government support for SWM installation. However, water utilities recently installed SWM in place of the traditional water meters due to the widely recognized benefits, such as convenient meter reading and accurate billing of water consumption. The United States has seen a surge in SWM installations even

though SWM implementation are not a requirement by law. In CA, a lot of water utilities had completed traditional metering, but are now in the process of converting to SWM [30]. Considering the advantages offered by SWM implementation, most water utilities have experienced several benefits from SWM implementation. Since most premises in CA had their traditional water meters installed from 1992, old traditional water meters required replacement [30]. Consequently, the USA has a positive situation where SWM implementation is increasing, signalling the acceptance of SWM technologies by the government, water utilities, and other relevant stakeholders.

### 3.2. United Kingdom

The diffusion of SWM in Europe has been slower in comparison to electricity and gas smart metering because the regulations (2012/27/EC) set by the E.U. concerning energy efficiency mainly enforce the implementation of smart energy metering for electricity and gas [32,33]. As presented in Figure 1b, 9 SWM projects conducted in the UK were reviewed in this study. Thames Water company has engaged in SWM implementation in England and Wales, reaching a milestone of 500,000 SWM and continuing on an upward implementations trajectory [34,35]. Southern Water made progress in SWM implementation at 92% completion with signs of continuing implementations [36]. The other SWM projects reviewed are considered as pilot/trial cases as they are below 1% completion, although most SWM projects displayed in Figure 1b were initiated in the past 10 years.

The water policies in England through the 'Water Industry Act of 1991' do not enforce SWM; thus, customers are not obliged to install a SWM in the UK, resulting in a slower progress of SWM implementation [37]. The UK government allows the optional installation of traditional water meters and SWM for several water companies (i.e., Affinity Water Anglian Water, Essex and Suffolk Water, South East Water, Southern Water, Sutton, East Surrey Water, and Thames Water) in water-stressed areas. Additionally, the water authorities (e.g., Defra and Ofwat) do not consider SWM as a critical and compulsory technology [38]. Scottish Water installed 3000 SWM in commercial properties to save water and to improve leak detection, promoting SWM implementation in this area [23]. However, the SWM diffusion rate in the UK remains relatively low, although many relevant water authorities make recommendations of SWM implementation.

### 3.3. Australia

Numerous SWM projects have been conducted in Australia [22]. Among them, 43 SWM projects have been reviewed, and several SWM implementations were presented in Figure 1c, with relatively high SWM completion percentages. The SWM implementation of Water Corporation of Western Australia (Kalgoorlie Smart Metering Trial) in Queensland, TasWater (Southern Water at the time of the project) in Tasmania, and City West Water in Victoria were completed in 2015 [39]. These SWM implementations in Australia have been used as early and successful SWM implementation examples. Between 2010 and 2012, the Kalgoorlie SWM project was implemented, and 13,800 units were installed for AUD 4 million [40]. Several achievements were noted in the Kalgoorlie SWM project, including a reduction in operating costs leading to savings of AUD 4.5 million per year, 13% reduction in overall water consumption, and improved leak detection [22,41].

In the Southern residential area, TasWater conducted an AUD 36 million SWM project with 46,000 units installed. From the results of this project performed in 2014, both a 37% reduction in costs to supply water and 10% reduction in water consumption were achieved [39]. Additionally, a strong media campaign was performed by the water utilities to ensure both the acceptance and awareness of the SWM implementations for residents, resulting in reduced customer complaints compared to those from the previous billing system using traditional water meters [23,39].

The water policies of Australia do not emphasize SWM implementations, but enforce adherence to 'AS4747 standards', which allow the installation of traditional water meters [9]. Although many water utilities have fully funded their SWM projects, external

funding, including government grants, generally encourage SWM implementation [42]. Since the benefits of SWM may be overridden by SWM implementation costs in the short term, this fact usually puts pressure on the water utilities, forcing them to abandon SWM project proposals and divert their attention to other issues (i.e., water and wastewater treatment projects). Based on the comprehensive review of the pilot/trial cases and SWM implementations conducted in Australia, a relatively slow SWM implementation trend after 2015 was perceived.

### 3.4. Israel

The water laws in Israel are very strict on water metering particularly based on the Law for 'Water Measurement of 1959' [43]. The government has supported various innovations in SWM by encouraging many Israeli venture companies to deal with water efficiency technologies [44]. Because Israel is a water-scarce country with long-term drought and possible threats to stable water supply, the early SWM implementations in Israel were driven by water scarcity issues and favorable government water policies [45]. As is evident by Figure 1d, some SWM implementation projects in Israel have reached a 100% SWM completion. For example, the City of Petak Tikva had 70,000 SWM installed and 7 other cities in Israel had at least 10,000 SWM each, reporting that at least 25% of traditional water meters from 2 million water meters were already replaced with SWM in 2010 [46]. Mei Raanana in Figure 1d was the first city to introduce SWM in 2000, and the city is now well-networked and uses the SWM system for billing, providing real-time water readings, and empowering water utilities to offer additional assistance and information to customers [45].

### 3.5. South Korea

The transition from traditional water meters to SWM in South Korea started in 2000 when the integration of water management issues with information and communication technology (ICT) commenced [46]. Recently, a three-step strategy for SWM implementations involving technology development, standardization, and a standard frame for the application of SWM implementations has been launched [47]. As many as 137 pilot/trial cases were conducted in many cities in South Korea, signaling the intention of the government to conduct more SWM implementations. Correspondingly, most water utilities have been engaged in pilot/trial cases to ascertain both the use and operation of SWM technologies before SWM implementation.

As displayed in Figure 1e, 146 SWM projects with 137 pilot/trial cases and 9 implementations have been conducted. SWM projects, such as Gochang City, Taean City, and Damyang City, have been reported with almost 100% completion. Since those cities have remote and small areas with less dense populations, SWM implementation was performed quickly. One recent project was Seosan Smart City starting in 2016 as a drought response countermeasure, and the city requested water authorities to integrate ICT technologies, such as SWM, wireless data transmission, and a decision-making system, to reduce water leakage rates [48,49]. Seosan Smart City installed 1550 SWM, 30 base stations, 9 sub-district metering areas, and monitoring systems, and the SWM project cost USD 0.4 million, financed by Seosan city and the central government through its drought budget. The city achieved a decrease in leakage and error of both inflow and outflow of distributing reservoir, and an increase in customer satisfaction [49]. Considering the number of successful SWM implementations conducted in South Korea to date, coupled with the ambitious completion goals of SWM implementations set by government and water authorities, more SWM implementations are thus expected to be conducted in the near future [48,49].

### 3.6. Comparison of SWM Implementations among Countries

The SWM implementation status among countries in this review is compared in Figure 1. The information gathered in this study proves that both status and progress of SWM implementations are quite different among countries. Since successful SWM implementations severely depend on both water policies and determination of both national

government and water providers [10], the absence of strong water policies and political support for SWM implementations resulted in the slower spread of SWM implementations in the UK compared to the other countries with significant progress in SWM implementations, as several cities in those countries have already reached 100% metering.

The status of the USA implementations reveals that several cities have completed SWM implementations. Since 2015, the cities in CA have increased the rate of SWM implementations because the financial support offered through the Federal Bureau of Reclamation has created a driving force for SWM implementations. However, SWM implementations in the UK were slow, except for the area served by the Southern Water Company that recorded 92% completion, followed by England and Wales (served by Thames Water) recorded as 21.41% [36]. However, the other projects in the UK were considered as pilot/trial cases since they are less than 1% metered. This was influenced by the UK government water policies, which do not consider SWM as a compulsory issue.

Australia has many SWM implementations with high metering percentages around 70%. Although progress is noticeable in Australia's water metering, the SWM diffusion rate after 2015 has slowed down in comparison to the early stages of the SWM implementations before 2015. In Israel, at least five cities have reached 100% completion of SWM implementations. In view of the water scarcity situation in Israel, more SWM implementations are projected in the whole country to save more water resources. After 2015, several completed SWM implementations have been observed in South Korea and more than 135 pilot/trial cases are anticipated to develop into quick SWM implementations in the near future.

## 4. Impact of Water Policies on the SWM Implementations

Various changes in water policies concerning accurate water metering have occurred in the selected countries over the years. Compared to fixed charging, volumetric charging measured by water meters is a fairer way of charging for water services, as volumetric charging incentivizes customers to reduce water consumption by decreasing their bills [9]. Over the past 30 years, the selected countries in this study have transitioned from a fixed charging to a volumetric charging regime composed of traditional water meters and SWM, as displayed in Figure 2. The USA experienced drought in 1992, which led to the introduction of the 'Metering rule for all new service connections (1992)', and this rule was expanded in 1996 to enforce the implementation of traditional water meters for all new water connections in drought-prone areas in the USA. In the same period, the UK introduced the 'Water Industry Act (1991)' and its subsequent amendment (i.e., 'Statutory Instrument 3442 (1999)'), which led to the metering of domestic water connections. The 'National Water Initiative (2004)' introduced several regulations to improve water management in Australia after a long drought throughout the 2000s, and traditional water meters were installed throughout the country at the instance of the 'National Water Initiative (2004)' [50].

In 1998, Israel experienced a severe drought leading to the establishment of a Parliamentary Investigation Committee in 2001, which introduced several reforms, such as 'The Municipal Water and Sewage Law (2001)' [43]. The establishment of these strong water policies (i.e., the committee, law, and regulations) coincided with the first SWM implementation in Mei Raanana, verifying the impact of indirect water policies intervention on SWM implementations. South Korea entered the era of 'Integrated Water Resources Management (2000)' [51], and the transition from traditional water meters to SWM started after the 'Water Supply and Waterworks Installation Act (2010)' was issued [48].

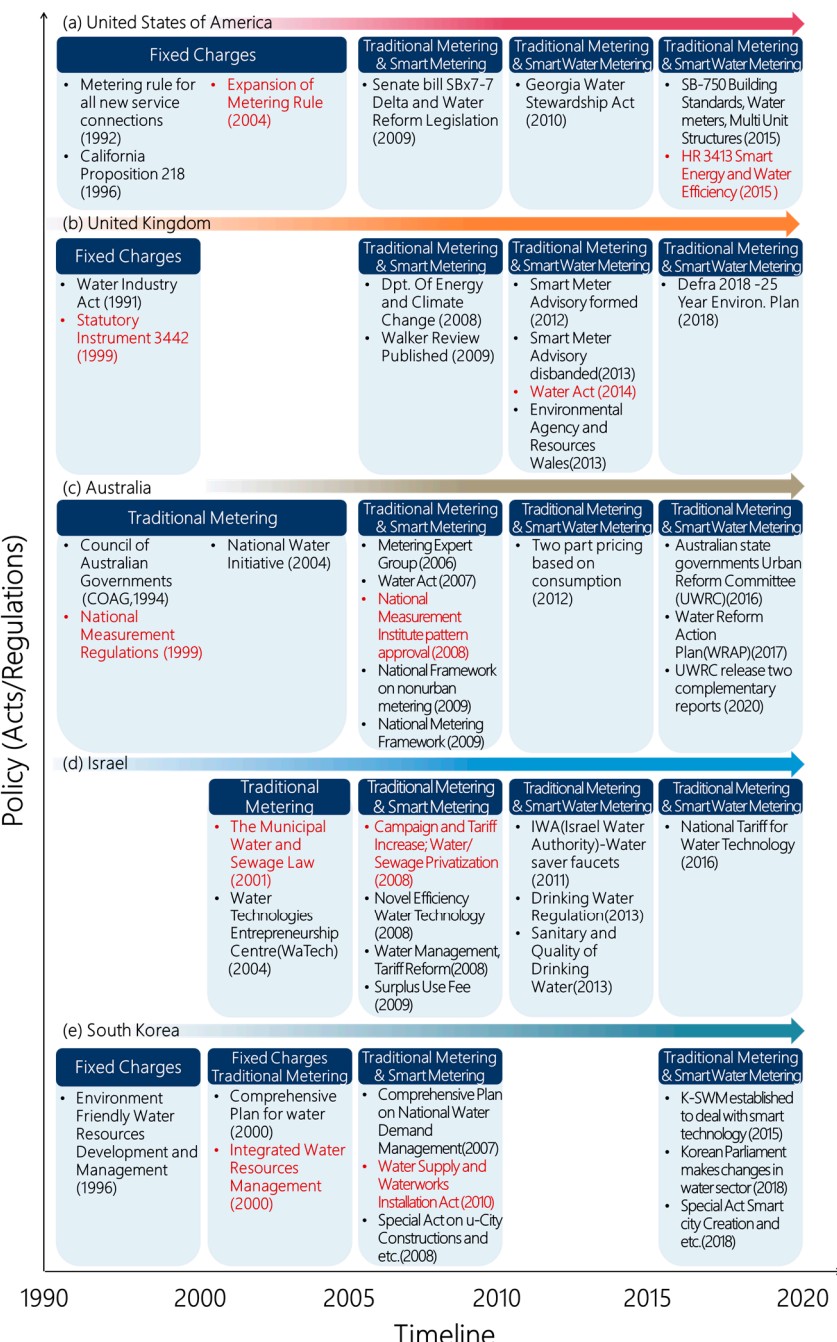

**Figure 2.** Evolution of water policies regarding tariff and regime of water metering in (**a**) United States of America, (**b**) United Kingdom, (**c**) Australia, (**d**) Israel, and (**e**) South Korea.

Considering that SWM implementations are complicated requiring the efforts of all stakeholders involved, governments through regulations and policies, SWM suppliers through equipment cost reductions, and customers through water use behaviors, the SWM implementation progress is complexly affected by numerous factors [9,52]. Over the years, governments across the world have been applying intensive water policies responding to water scarcity, population growth, and water demand management [53]. Direct water policy interventions specify the SWM installations whereas indirect water policy interventions promote SWM implementations unintentionally, such as water efficiency, water use reduction targets, drought relief plans, and water supply tax [52,53]. Since the improvement of water governance and policy is key to finding a solution to water insecurity [54,55], the water

policies of the selected countries and the impact of water policies on the implementations of SWM were analyzed.

### 4.1. United States of America

The national-level policy on water management in the USA is not coherent, thus water policies regarding smart water management have been developed at state and local levels [26]. Most water policies to enhance water accounting, reporting, and efficiency have supported SWM implementation [56]. For example, several laws and regulations from different states (i.e., Alabama, Arizona, California, and Washington) concerning both installation and usage of water meters generally require the billing of all the water supplied to customers based on metered water volume [57]. However, the water utilities still charge flat rates for water consumption in the case of public and private fire protection and water for street sprinkling among many other public services [57].

As presented in Figure 2a, CA introduced a law requiring the metering of all new water service connections in 1992. In 2004, the law was expanded to require all urban water suppliers to install water meters on their water service connections [28,57]. Since 2004, water suppliers from the Federal Central Valley project had to meet the metering requirement for all water service connections by January 2013 [26]. 'Senate bill SB x7-7 Delta reform legislation' issued in 2009 enforced water use reduction by 20% until 2020 [26,58].

According to 'California Proposition 218 (1996)', an increase in water service charges must be based on an increase in the costs. Thus, water utilities aimed at reducing costs should minimize the errors in measurement using SWM, decreasing water losses, preventing theft, and replacing the pipes [58]. However, from 2005 to 2010, traditional metering was dominant, therefore, the 'Georgia water Stewardship Act' was issued to encourage efficient water use in 2010. Finally, 'Smart Energy and Water Efficiency Act (HR 3413)' was passed in 2015, initiating five SWM pilot projects in different cities with an emphasis on leak detection and remediation [54] and intensifying SWM implementation after 2015, as is evident in Figure 2a.

The dynamic inclusion of all stakeholders has resulted in a rush in SWM implementations in the USA. As a direct example of strong water policy, the 'Government-Federal Bureau of Reclamation through WaterSmart Grants' has engaged water utilities in the installation of SWM in the City of Santa Ana, CA, by offering 'Water and Energy Efficiency Grants' for FY 2020 [30]. WaterSmart Grants distributed support grants to water utilities expressing interest in SWM projects, resulting in more SWM implementations. Through this government initiative in CA, several water utilities have now reached 100% SWM installation [30].

As an indirect example of water policies, the recurring drought conditions in CA inclined the spread of SWM implementations by requesting the Metropolitan Water Department to reduce water deliveries to utilities and to enforce 15% water share reductions [30]. This led to the City of Santa Ana declaring a water supply shortage, thereby requesting water customers to reduce their water use by 12% relative to the base year (2013), ensuing in an overall 16.97% water use reduction over nine months [30]. Although the drought conditions were improved in 2016, most water utilities in CA decided to upgrade small SWM pilot/trial cases to advanced water demand management projects through intensive SWM implementations [59,60].

### 4.2. United Kingdom

The national water policies of the UK are based on a mix of metered charging or fixed charging based on the ratable value of the properties [61]. Whereas 37% of households in England were water-metered in 2010 based on Pike research (2010), 50% of households in England were water-metered by 2018 based on the Environmental Agency of the UK (2018), indicating the slow progress in the diffusion of water metering in the UK Generally, the adoption of SWM has not been given priority in the UK, and Ofwat projected a 90% metering target by 2030, irrespective of the type of water metering technology [54]. Although

several areas in England and Wales are under water stress, especially the South East of England [26,37,54], water metering is not compulsory in the UK, except for circumstances described in 'Water Industry (prescribed conditions) Regulations' [62].

As indicated by Figure 2b, the impact of water policies on change in the type of water metering is observed, while traditional metering implementation and SWM has been increasing in the UK. The 'Water Industry Act' was passed in 1991 and then amended ('Statutory instrument 3442') in 1999 by differentiating non-compulsory domestic water metering from compulsory domestic water metering [20]. In 2008, the 'Department for Environmental Food and Rural Affairs (DEFRA)' commissioned a review to examine the tariff system for water and sewage. Then, the 'Walker review' was published in 2009, proposing more systematic water metering to achieve efficient water use [63]. Additionally, Ofwat acknowledged the potential for SWM by recognizing its role in leakage detection; however, Ofwat did not explicitly recommend SWM implementation.

Based on the recommendations from the Environmental Agency and DEFRA, the 'Smart Meter Advisory Group' was set up and operated in 2012. However, the Smart Meter Advisory Group was disbanded in 2013. As a result, no government water policy compelled water companies to implement SWM [54]. In 2014, the 'Water Act' was introduced to encourage the implementation of water metering for the customer's benefit [64]. Without specifying the detailed functions of SWM to produce granular data, Ofwat revealed a strategy requiring the use of customer data for improved customer service and communications in 2014 [54]. In 2018, DEFRA published an environmental plan stating both needs and measures to have a personal water consumption target in England and the National Infrastructure Commission (NIC) recommended SWM implementations to improve water efficiency [38,65]. In 2019, the NIC, the Committee on Climate Change, and the House of Commons Environment Food and Rural Affairs (EFRA) Committee recommended the government to enforce compulsory water metering in the UK. However, the UK government still maintained a non-compulsory water metering policy in parliament [37].

### 4.3. Australia

The reforms agreed by the 'Council of Australian Governments (COAG)' in 1994 produced initiatives that have continued to make remarkable progress towards sustainable water management over the past 27 years [66]. The 'National Water Initiative (NWI)' was implemented in 2004 through the COAG Agreement, and all parties involved in the agreement committed to the terms of the initiative by adhering to both objectives and actions to increase water use efficiency [67]. In 2005, the review of the NWI commenced under the guidance of COAG, the National Water Commission (NWC) and the Natural Resource Management Ministerial Council (NRMMC) published progress reports of NWI every two years [67]. Figure 2c presents a summary of the acts and regulations that have occurred in Australian water metering since the 1990.

The 'Water Act (2007)' had the purpose of compiling, maintaining, and issuing appropriate national water standards to achieve effective water meter implementations since there was no Australian standard for water meters or ancillary data collection systems [68]. Thus, universal and accurate water meters following Australian standards were required to secure the integrity of the water management system and to develop better water planning and allocation based on reliable water consumption information. Accordingly, the National Metering Expert Group was established in 2006 to work in conjunction with the National Measurement Institute to develop metering standards. Then, urban water meter implementations satisfying the requirements of the 'National Metering Framework (2009)' commenced in several states (New South Wales, Queensland, Victoria, South Australia, Western Australia, Tasmania, Northern Territory, and Australian Capital Territory) from 2009 to 2012 [69]. Since most metering implementation plans were completed by 2011, the aforementioned states introduced two-part tariffs (i.e., metered water charge and fixed service fee component) for urban water replacing fixed water charges based on property values and 'free allowances' [70]. Most Australian capital cities and many

urban areas adopted two-part tariffs for potable water supply. For example, Tasmania implemented water meters across all urban areas and then effected volumetric charging from July 2012 [50].

COAG developed the 'National Framework for non-urban water metering' to establish a national standard for non-urban water meters, and to improve both accuracy and extent of water metering [56]. After July 2010, 'National Framework for non-urban water metering' came into effect, enforcing meters to be pattern-approved, to have the capacity for telemetry, and to be installed in accordance with ATS4747 [71]. The NWC was abolished in 2014 and the Australian state governments through the 'Urban Water Reform Committee (UWRC)' have been working to reform 'National Metering Framework' since 2016. Subsequently, the 'Water Reform Action Plan (WRAP)' was released in 2017. The WRAP described the requirements for customers subject to water metering, the water customers exempt from metering, and the equipment used on all water supply work [72]. WRAP also described the new requirements for technicians who fit meters, telemetry, record-keeping and reporting rules, and the procedure [73]. Some parts of the regulations which relate to new and replacement meters, faulty meters, and inactive meters commenced on 1 April 2019 [73].

*4.4. Israel*

In Israel, persistent years of drought have been countered with strict water policies leading to the significant increase in water tariffs and an additional surplus fee on excessive water consumption. These regulations gave rise to many water technology ventures and the implementation of water technologies at all scales from water-saving devices, SWM and desalination plants [43]. Israeli water sector implemented nine key innovations in its water policies. These innovations issued by the government directly affected the implementation of water-efficient technologies by establishing a remarkable relationship between the industry–utility–university ecosystems. The economic incentives offered by the government to reduce water demand in both the urban and agricultural sectors have emerged as the main drivers for the development of innovative water management devices, such as SWM [44].

Israel is a water-scarce country, and persistent drought and threats to its water security led to the enactment of the 'Water Law' in 1955, which states that "a man shall not supply water unless it is metered" [43]. The Israeli government enacted water policies to keep up with water-related innovations as evidenced in Figure 2d, coupled with strong water pricing policies (i.e., 'tariff reform (2008)' and 'Surplus use fee (2009)') to encourage the use of water-saving technologies. A drought occurred in 1998 and resulted in acute water shortages. As displayed in Figure 2d, the 'Municipal Water and Sewage Law' was then enacted in 2001, which ring-fenced municipal water services. A parliamentary investigation committee for the water sector was consequently established in 2002, leading to the generation of several water policy reforms for over 15 years [44]. In the same period, an inter-ministerial committee in Mekorot city established a 'Water Technologies Entrepreneurship Center (WaTech)' in 2004 to support business ventures, starting 35 SWM pilot/trial cases from 2004 to 2007. In 2006, a successful water-saving campaign was launched, then expanded in 2008 as part of the eighth innovation. Additionally, the 'Novel Efficiency Water Technology (NEWTech)' program where 26 government-funded water technology start-ups managed to attract USD 700 million from private investors was established in 2008.

In the period 2010–2011, the Israeli Water Authority was engaged in a campaign where toilet water saver faucets were distributed after finding that 35% of overall household water consumption was toilet flushing [45]. Thus, the use of such devices would reduce water consumption by around 30% according to IWA (Israeli Water Authorities) estimates. Several policies, such as 'Drinking water regulations' (2013) and 'Sanitary and quality of drinking water' (2013) [45], enforced in Israel have generated more SWM implementations either directly or indirectly. For example, water loss fines encouraged the development of both water loss detection and dynamic water pressure equipment [74]. Additionally, Mei Raanana started using the SWM system in 2002 as a token of coherence with the

empowering of municipal services and establishment of a 'Parliamentary Investigation Committee' [45]. In 2016, a change in national tariff for portable water was established to pursue water demand management, while ensuring that everyone has access to life-line consumption volume at an affordable price [75].

### 4.5. South Korea

South Korea's water policy development started in the 1960s [48,76]. The Korean water policies have experienced a change from a supply oriented quantitative approach to a qualitative approach focusing on water quality, ecology, and the environment over the years. In 1996, the 'Environmentally friendly Water Resources Development and Management Plan' was established to develop multipurpose dams [48]. Both the 2000 and 2007 'Comprehensive Water Plan' resulted in water savings of 930 million tons of tap water from the expansion of water services and the replacement of aging infrastructures [26]. Since then, Korean water authorities were looking for alternative options to address water scarcity using available water and assets, while minimizing or deferring capital expenditures.

A distinctive approach to SWM in South Korea was initiated in 2000 to achieve effective goals with 'Integrated Water Resources Management (IWRM)' for future water issues. In 2010, the shift in the water policies has concentrated on tackling climate change in concert with sustainable water management by issuing 'Water Supply and Waterworks Installation Acts (2010)' [26,48]. In 2018, the Korean parliament made huge changes in water management initiating a paradigm shift, where real-time monitoring and remote controlling systems were setup using ICT and AI, including SWM implementations. SWM is well in line with South Korea's national policies towards a creative economy aiming to create jobs from innovation combined with information and communication technology [47]. South Korea has also set up the Environmental Technology Incubator in Korea to support venture capital and provide incentives for SWM technology developments [26]. Correspondingly, the government has been involved in SWM implementations as in the case of Seosan city, where the water utilities were supported by the central government in a USD 0.4 million project through the government drought relief budget.

### 4.6. Impact of Water Policies on the Transition from Trials/Pilot Cases to Implementations of SWM

During the last three decades, a considerable number of SWM pilot/trial cases have been conducted all over the world [77]. During this period, the transition from fixed water charges to metered water charges has occurred (see Figure 2), and recent digital evolution resulted in the spread of SWM [9,20,28]. Many countries have adopted SWM technologies to improve water efficiency. Generally, SWM pilot/trial cases started in the 1990s and are being conducted to date. Due to several reasons, such as lack of strong government water policies [78], high costs of equipment [79,80], and lack of expertise [81], the transition period from pilot/trial cases to field implementations of SWM among countries is quite different, as displayed in Figure 3. In this study, the impact of water policies on the spread of SWM implementations was investigated. Although more SWM projects may have been conducted in the selected countries, the investigation performed in this study is deemed to be sufficient to draw a correlation between water policies and the spread of SWM implementations.

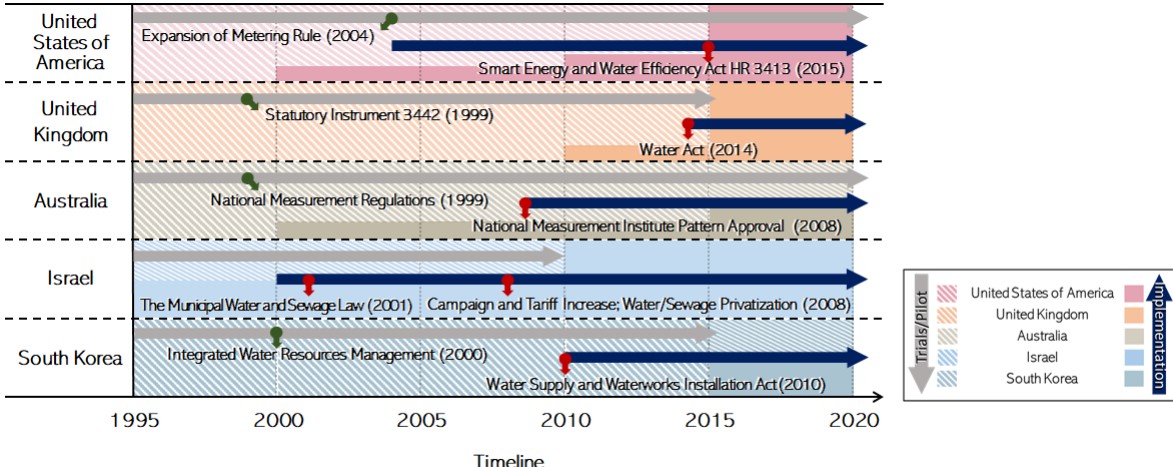

**Figure 3.** Major water policies impacting the implementation of smart water metering in selected countries.

Overall, no compulsory SWM policy was found in most countries in this study. However, since numerous benefits exist from SWM implementations beyond metering and billing [14,23,82], many water utilities have been adopting SWM implementation as water demand management techniques. Generally, SWM implementation has impacted the operations of utilities positively, according to the previous studies, presenting the benefits derived from the SWM implementations [27,83]. The evolution of SWM from 1990 in the selected countries has been presented in Figure 3. Although several changes in water policies have occurred since 1990, there is no compulsory law for SWM in the selected countries. However, the adoption of SWM is slowly increasing with time as the proportion of pilot/trial projects decreases compared to SWM implementations, as is evident by Figure 3. Between 1995 and 2010, pilot/trial cases for SWM were dominant as these were preliminary stages of SWM technologies. Additionally, during this period, most countries in this review were mainly engaged in traditional metering. Most countries did not experience intensive SWM implementations because they were conducting pilot/trial cases prior to SWM implementations. After 2010, SWM implementations kept increasing and all countries experienced more concentrated SWM implementations, despite the variances in both endpoints and completion of SWM implementations depending on the water policies (i.e., acts and regulations) enacted to encourage SWM implementations.

In the USA, the 'Expansion of Metering Rule' in 2004 expanded the metering era by requiring all urban water suppliers in CA with at least 3000 customers to install water meters on all service connections [57]. In CA, the 'Senate bill SB x7-7 Delta and Water legislation (2009)' paved the way for SWM implementations by requiring new water demand management measures, technologies, and approaches coupled with a 20% reduction in per capita water use target. The launching of the 'Smart Energy and Water Efficiency Act HR 3413 (2015)' further encouraged the use of water-efficient technologies and resulted in the increased SWM implementations. The water policies of the UK do not specifically encourage SWM implementation, while the issuing of 'Water Industry Act Statutory Instrument 3442 (1999)' initiated the implementation of traditional meters in drought-prone areas. The Thames and Southern Water Companies in the UK have engaged in SWM implementations in their service areas, irrespective of non-compulsory SWM regulations from the national water policies. The 'Water Act (2014)' was neutral on SWM implementations, but encouraged water saving and efficiency. Thus, the 'Water Act (2014)' provided a basis for implementing SWM to the water companies intending on utilizing numerous benefits of the SWM technologies.

Since 1999, the 'National Measurement Regulations (1999)' launched in Australia have required the standardization of all metering practices to achieve a universal measurement method [84]. These requirements could be attained using traditional meters and encouraged their implementations. The 'Pattern approval regulations by the National Measurement

Institute' presented in 2008 required all water meters to be installed according to standard AS4747 demanding telemetry ability, leading to numerous SWM implementations since 2010 [85]. For Israel, most pilot/trial cases for SWM were performed before 2010, and SWM implementation started in 2002 after the establishment of the 'The Municipal Water and Sewage Law (2001)' and the subsequent issuing of both acts and regulations to encourage water saving and the use of water efficiency technologies. Successful campaign and tariff increase policies were initiated in 2008, hence, water consumption was greatly reduced by implementing SWM and water-saving devices. In South Korea, SWM pilot/trial cases have been encouraged by the initiation of 'Integrated Water Resources Management' policy since 2000. Most SWM implementations surfaced from 2010 onwards as the 'Water Supply and Waterworks Installation Act (2010)' was passed to support water demand management technologies [48]. Recently, integrating ICT into SWM implementations in South Korea brought innovative water management and SWM implementations to improve water security and welfare by addressing global water issues [47].

In conclusion, the issuing of water policies and supporting grants for sustainable water management by the governments is critical to spread SWM implementations in the countries evaluated in this study. Most SWM implementations that have surfaced in the analyzed countries have been initiated after water policy reforms were established. Thus, more SWM implementations were conducted as the robust water policies encouraged the use of water efficiency equipment, therefore, broadening the supplier base, reducing the cost of SWM technologies, and improving the validity of business cases for SWM implementations.

## 5. Market Status and Forecast

Generally, the global market for SWM has grown since the inception of SWM technologies in the 1990s. A report prepared by IMARC group presented the value of the global SWM market as USD 5.92 billion in 2020 [86]. The infrastructural investment by governments of developed countries in the creation of smart cities has generated a demand for effective water demand management by encouraging homes and businesses to use SWM technologies. The increase in the SWM market has also been encouraged by the easy availability of SWM technologies, the rising popularity of the Internet of Things (IoT)-based devices, and the development of smart cities across the globe [87]. The growth in demand for SWM equipment is expected to further increase in the future due to the development of low power wide area (LPWA) network technologies, providing enhanced coverage and connectivity for SWM [88]. The application of consistent water policies to optimize the use of water resources and to promote sustainable development is expected to drive the SWM market further [53,89].

The USA has contributed significantly to the growth of the global SWM market over the years. As presented in Figure 4, the SWM implementations in the USA began around 2004 as the USA state governments enforced laws that required reduction in water use and encouraged the use of water-efficient equipment (see Figure 3). In 2020, the global SWM market share for North America was 57.3% with the USA contributing a 91.1% share valued at USD 3.09 billion. As displayed in Figure 4, regarding the SWM market growth of the USA from 2009 to 2026, the market contribution of the USA has grown gradually from USD 421 million in 2009 to USD 3.09 billion in 2020 at a CAGR rate of 9.7% and is expected to grow to USD 5.34 billion by 2026 [46,86]. The SWM market growth in the USA is attributable to the water policies of various state governments aiming to improve water efficiency in the country. The increase in SWM implementations of the USA from 2015 is synonymous with the introduction of "HR 3413 Smart Energy and Water Efficiency Act" coupled with water efficiency grants from the Federal Bureau of Reclamation, which encouraged utilities to reduce water consumption and to improve water demand management.

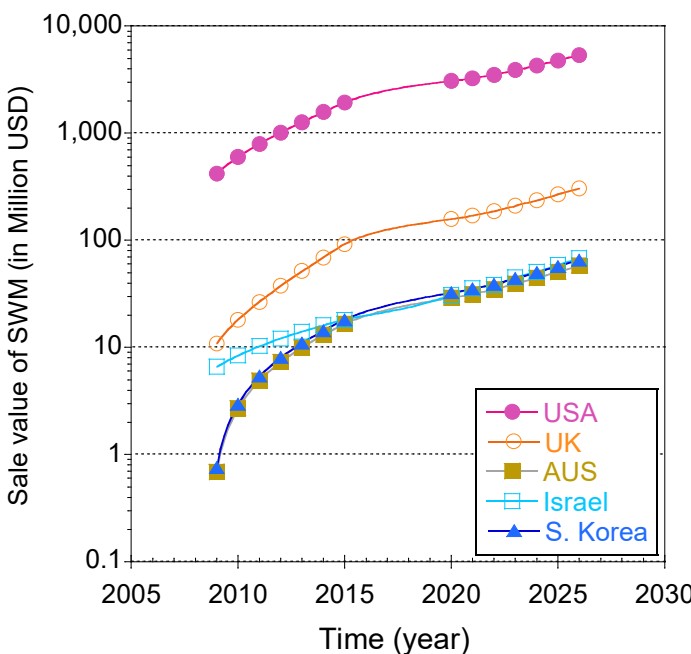

**Figure 4.** Market status and forecast of sale values (in million USD) of smart water metering with time in the selected countries.

The global market share for SWM for Europe in 2020 was 23.7% valued at USD 1.40 billion [89]. The UK contributed a market share of 11.3% valued at USD 158.4 million with a CAGR of 11.4% in the period 2015–2020. The sales value for SWM in the UK has grown from USD 11 million in 2015 to USD 158.4 million USD in 2020, and it is expected to reach USD 306.2 million by 2026. The UK water policy is neutral on SWM implementations; therefore, traditional metering is dominant up to date. As indicated by Figure 4, significant growth followed by steady growth in the SWM market is observed due to the limited adoption of SWM by water companies only in drought-prone areas of the UK. Since the "Water Act of 2014" emphasized the improvement in water efficiency focusing on drought-prone regions, the SWM implementations in the UK remained slow in 2018 as only 50% was metered [37]. The transition to SWM may remain slow, unless the government introduces robust water policies encouraging SWM implementation.

As presented in Figure 4, the SWM market in Australia reached USD 29.3 million in 2020 with a CAGR of 11.8% since 2015. Australia accounted for 4.5% of the Asia-Pacific's contribution to the global SWM market from 2015–2020 [86]. The market expanded from USD 0.69 million in 2009 to USD 29.3 million in 2020 and is forecast to reach USD 58.2 million by 2026. After the introduction of the 'National Measurement Institute Pattern Approval regulation (2008)' encouraged SWM implementations [90], there was a surge in the demand for SWM as many water utilities engaged in pilot/trial cases and implementations. Since two-part pricing policy based on water consumption was adopted in 2012, the quest for more efficient ways of collecting meter readings and billing caused the widespread of SWM implementations with real-time water consumption.

Figure 4 also showed that the market share of Israel was greater than that of both South Korea and Australia in 2009–2015. Additionally, the SWM market of Israel increased to around USD 31 million by 2020 and is expected to reach USD 68 million in 2026. This SWM market trend of Israel is synonymous with the findings in Figure 3, showing that Israel's SWM implementations started in 2002 in Mei Raanana and seven other cities that were initiated by 'Municipal Water and Sewage Law' in 2002 [46]. Generally, the SWM market of Israel has continued to grow because of the increased support of various SWM technologies from consistent and robust water policies of Israel. In particular, the 'Campaign and Water Tariff Increase (2008)' encouraged the use of water saving technologies to reduce water consumption, therefore growing the SWM market of Israel [43].

As displayed in Figure 4, the SWM market in South Korea reached USD 32.6 million in 2020 for a CAGR of 12.3% during 2015–2020, representing a great increase in the period 2015–2020. During this time, pilot/trial cases were conducted concurrently with several SWM implementations. It is evident in Figure 3 that most SWM implementations were conducted in the period 2010–2020 after the 'Water Supply and Waterworks Installation Acts' were issued [51]. The SWM market of South Korea is predicted to grow to USD 65.7 million by 2026, and the SWM market may double based on the number (i.e., $n = 137$) of the pilot/trial cases currently being conducted. A 'Special Act (Act on u-City Constructions and etc., 2008) on smart cities' was enacted in South Korea in 2008 then revised (Act on Smart City creation and etc., 2018) in 2018 to approach smart cities as a platform. South Korea had 84 smart city projects in place by 2019, the development of smart cities contributed to the rise in the demand for SWM technologies consequently increasing the SWM market [91].

## 6. Conclusions

This review paper mainly analyses both the status and progress in SWM implementations in selected countries and focuses on reviewing the transition from pilot/trial cases to SWM implementations. Data were collected by reviewing water policies, journals, news articles, utility websites, and close communication with stakeholders involved. Despite the countless benefits that can be derived through SWM implementations, the diffusion of SWM technologies has been slow due to various challenges, including lack of support from customers, weak cost–benefit analysis, lack of expertise, and absence of supporting water policies.

Based on this comprehensive review, the selected countries in this review have transitioned from a fixed charging to a volumetric charging regime composed of traditional water meters and SWM over the past 30 years. Although governments across the world have been applying water policies responding to water scarcity, population growth, and water demand management, both the status and progress of SWM implementations are quite different among countries. Since successful SWM implementations severely depend on both water policies and the determination of both national government and water providers, the absence of strong water policies and political support for SWM implementations resulted in the slow spread of SWM implementations. Although several changes in water policies have occurred since 1990, there is no compulsory law for SWM. However, the adoption of SWM is slowly increasing with time as the proportion of pilot/trial projects decreases compared to SWM implementations. Between 1995 and 2010, pilot/trial cases for SWM were dominant as these were preliminary stages of SWM technologies. During this period, most countries in this review were mainly engaged in traditional metering and did not experience intensive SWM implementations because they were conducting pilot/trial cases prior to SWM implementations. After 2010, the number of SWM implementations kept increasing and all countries experienced more concentrated SWM implementations, despite the variances in both endpoints and completion of SWM implementations depending on the water policies (i.e., acts and regulations) enacted to encourage SWM implementations. The global market for SWM has consistently grown since the inception of SWM technologies in the 1990s, presenting a global SWM market value of USD 5.92 billion in 2020. Finally, the application of consistent and robust water policies to optimize the use of water resources and to promote sustainable development is expected to drive the SWM market further. For future study, the impact of SWM implementation on the sales of water utilities, the reduction in water consumption, and the mitigation of greenhouse gas emissions needs to be investigated.

**Author Contributions:** Methodology, S.M., H.W.G., S.Y.J. and D.H.L.; validation, J.M.L. and S.L.; investigation, S.M. and J.C.J.; resources, S.M. and J.C.J.; data curation, S.M., D.J.L., H.W.G., S.Y.J. and D.H.L.; writing—original draft preparation, S.M. and J.C.J.; writing—review and editing, J.C.J., J.M.L. and S.L.; visualization, H.W.G., S.Y.J. and D.H.L.; project administration, J.C.J., J.S.C. and J.M.L.;

funding acquisition, J.M.L. and S.L. All authors have read and agreed to the published version of the manuscript.

**Funding:** This research was funded by Korea Ministry of Environment (MOE) (2019002950003).

**Institutional Review Board Statement:** Not applicable.

**Informed Consent Statement:** Not applicable.

**Data Availability Statement:** The data is available on the request from the corresponding author.

**Acknowledgments:** This work was supported by Korea Environment Industry & Technology Institute (KEITI) through Intelligent Management Program for Urban Water Resources Project.

**Conflicts of Interest:** The authors declare no conflict of interest.

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
