# Peer review of "Role of Water Policies in the Adoption of Smart Water Metering and the Future Market"

_water, doi:10.3390/w14050826_

Round 1

Reviewer 1 Report

Dear authors,

the paper is well written, theliterature review is really interesting but I have one suggestion :

 - Please define a way to assess the impact of smart water metering implementation on the sales of water utilities and how the implementation of metering systems influence water consumption and the possible pressure on water ressources ? The paper still general and aboslutely not accurate to estimate cost-benefit of implementing SWM ...when you present the growth of metering market don't forget that meters are paid par consumers so if the regulator decide to implement meters it's easy to know the number of average meters to implement...The presented study is not really consistent beceause the estimation onf potential market is easy to estimate based on the size f water utilities and the population delivred... You should propose a significant and consistent way to asses the impacts of SWM on water utilities as a real scientific contriubtion of your work otherwise you paper is an interesting literature review without a real added value.

Author Response

Responses to Comments by Reviewer #1

Comment 1-1

This paper is well written, the literature review is really interesting, but I have one suggestion:

Please define a way to assess the impact of smart water metering implementation on the sales of water utilities and how the implementation of metering systems influences water consumption and the possible pressure on water resources? The paper still general and absolutely not accurate to estimate cost-benefit of implementing SWM ...when you present the growth of metering market don't forget that meters are paid par consumers so if the regulator decide to implement meters it's easy to know the number of average meters to implement.

Response 1-1

The authors appreciate the reviewer’s insightful comments. The authors definitely agree with reviewer’s comments. However, some of the suggestions (e.g., impact of smart water metering on the sales of water utilities and water consumption) from reviewer 1 are out of scope of this study. Actually, the impact of smart water metering implementation on water consumption is intensely analyzed for our future study and the subsequent paper is in progress. Thus, the authors have warranted the future study regarding the impact of smart water metering implementation on the sales of water utilities and water consumption in conclusions section (see lines 586-587).

“For future study, the impact of SWM implementation on the sales of water utilities, the reduction of water consumption, and the mitigation of greenhouse gas emission needs to be investigated.”

Comment 1-2

The presented study is not really consistent because the estimation of potential market is easy to estimate based on the size of water utilities and the population delivered... You should propose a significant and consistent way to assess the impacts of SWM on water utilities as a real scientific contribution of your work otherwise your paper is an interesting literature review without a real added value.

Response 1-2

Similar to the reply to comment 1-1, the authors definitely agree with reviewer’s comments. However, some of the suggestions (e.g., impact of smart water metering on the sales of water utilities and water consumption) from reviewer 1 are out of scope of this study. Actually, the impact of smart water metering implementation on water consumption is intensely analyzed for our future study and the subsequent paper is in progress. Thus, the authors have warranted the future study regarding the impact of smart water metering implementation on the sales of water utilities and water consumption in conclusions section (see lines 586-587).

“For future study, the impact of SWM implementation on the sales of water utilities, the reduction of water consumption, and the mitigation of greenhouse gas emission needs to be investigated.”

Reviewer 2 Report

This is an important topic and the authors present considerable data on the growing use of smart water meters in different countries.  This in itself is a worthwhile contribution and the paper should be published.  It would be an even more significant contribution if the authors systematically had fleshed out in more detail the explanations for differences between countries.  They mention that policies mandating the use of smart meters will lead to more being installed and they provide lots of anecdotal evidence, but more exploration of the factors responsible for the trends they observe would be welcome.  One stylistic note: the preposition "of" is inserted inappropriate and unnecessarily in several sentences (e.g., in the abstract: "Despite of the countless benefits through the implementations of SWM").

Author Response

Comment 2-1

This is an important topic, and the authors present considerable.  This is a worthwhile contribution, and the paper should be published.  It would be an even more significant contribution if the authors systematically had fleshed out in more detail the explanations for differences between countries.  They mention that policies mandating the use of smart meters will lead to more being installed and they provide lots of anecdotal evidence, but more exploration of the factors responsible for the trends they observe would be welcome.  

Response 2-1

The authors appreciate the reviewer’s perceptive comments. The authors agree with reviewer’s comments and add more factors to explain the different status in SWM among selected countries (see lines 86-102, 113-115, 128, 166-168, 171, and 186-188).

The authors highlighted some factors that influence the implementation of SWM in the manuscript.

“The benefits associated with SWM are discussed in the mentioned review paper by Ian Monks et al. Challenges/Factors such as privacy and security, data management, technical capacity among others are mentioned as limitations to the implementation of SWM. Other factors such as high cost of equipment, lack of standard and common implementation protocols were highlighted as hindrances to implementation. This study was then structured to investigate only the impact of water policy on SWM implementation (see lines 87-104).”

“The issuing of government grants to water utilities is discussed as a factor assisting the implementation of SWM (see lines 114-117).”

“Factors such as convenient meter reading, accurate billing of water consumption are mentioned (see line 129-131).”

“Several factors affecting SWM implementations were discussed here, for example the reduction in operating costs, reduction in overall water consumption, and improved leak detection (see lines 169-171).”

“The use of strong media campaigns by utilities to ensure acceptance and awareness of SWM implementations was considered as a factor affecting SWM implementations (see lines 174).”

“Water scarcity is given as driver for SWM implementations (see lines 189-191).”  

“Considering that SWM implementations are complicated requiring the efforts of all stakeholders involved, governments through regulations and policies, SWM suppliers through equipment cost reductions, and customers through water use behaviours, the SWM implementation progress is complexly affected by numerous factors [9,55]. Over the years, governments across the world have been applying intensive water policies responding to water scarcity, population growth, and water demand management [56]. Direct water policy interventions specify the SWM installations whereas indirect water policy interventions promote SWM implementations unintentionally such as water efficiency, water use reduction targets, drought relief plans, and water supply tax [55,56]. Since the improvement of water governance and policy is key to finding a solution to water insecurity [57,58], the water policies of the selected countries and the impact of water policies on the implementations of SWM were analysed (see lines 262-271). ”

Comment 2-2

One stylistic note: the preposition "of" is inserted inappropriate and unnecessarily in several sentences (e.g., in the abstract: "Despite of the countless benefits through the implementations of SWM").

Response 2-2

The authors appreciate the reviewer’s editorial comments. The authors agree with reviewer’s comments, and revise the whole manuscript to correct the use of preposition.
